# The Role of Oxidative Stress in Manganese Neurotoxicity: A Literature Review Focused on Contributions Made by Professor Michael Aschner

**DOI:** 10.3390/biom13081176

**Published:** 2023-07-28

**Authors:** David C. Dorman

**Affiliations:** Department of Molecular Biomedical Sciences, College of Veterinary Medicine, North Carolina State University, 1052 William Moore Dr, Raleigh, NC 27606, USA; david_dorman@ncsu.edu

**Keywords:** manganese, neurotoxicity, oxidative stress, mechanisms, *C. elegans*

## Abstract

This literature review focuses on the evidence implicating oxidative stress in the pathogenesis of manganese neurotoxicity. This review is not intended to be a systematic review of the relevant toxicologic literature. Instead, in keeping with the spirit of this special journal issue, this review highlights contributions made by Professor Michael Aschner’s laboratory in this field of study. Over the past two decades, his laboratory has made significant contributions to our scientific understanding of cellular responses that occur both in vitro and in vivo following manganese exposure. These studies have identified molecular targets of manganese toxicity and their respective roles in mitochondrial dysfunction, inflammation, and cytotoxicity. Other studies have focused on the critical role astrocytes play in manganese neurotoxicity. Recent studies from his laboratory have used *C. elegans* to discover new facets of manganese-induced neurotoxicity. Collectively, his body of work has dramatically advanced the field and presents broader implications beyond metal toxicology.

## 1. Introduction

Manganese is an essential nutrient that plays a critical role in protein, lipid, and carbohydrate metabolism in animals and humans. Manganese serves as an enzyme cofactor and is incorporated into several metalloenzymes, including manganese superoxide dismutase (MnSOD), arginase, glutamine synthetase, phosphoenolpyruvate decarboxylase, and pyruvate carboxylase [1,2]. Mammalian tissues normally contain 0.3–3.0 μg Mn/g wet tissue weight [3]. The body’s nutritional requirements for manganese are normally met through dietary intake via food and drinking water. The estimated safe and adequate daily dietary intakes (ESADDI) for manganese required to maintain body stores is 2 to 5 mg/day in adults and 1.5 to 2.0 mg/day for children 4 to 6 years of age [4]. To ensure adequate nutrition in neonates, manganese is often added to infant formula because there is a greater need for this element during growth and development [5].

Exposure to excessive amount of manganese can result in manganese neurotoxicity, producing adverse effects most notably in the human extrapyramidal system [6,7,8]. Neurotoxicity can occur following high-dose oral, inhalation, or parenteral exposure to manganese. The development of neurotoxicity following different routes of exposure indicates that the dose to target tissue is the critical determinant of manganese toxicity, regardless of route. This association between manganese and neurotoxicity was first noted by Couper in 1837 who reported abnormal neurologic effects in workers at an ore-grinding plant where “black oxide of manganese” was processed [9]. Most epidemiologic research on manganese conducted during the late 20th century focused on occupational inhalation exposure. Subsequent epidemiologic studies of welders, manganese miners, battery producers, and other manganese workers have clearly established a causal association between chronic high-dose-manganese exposure via inhalation and neurotoxicity [10]. Hallmarks of manganese neurotoxicity in adults include behavioral changes, cognitive deficits, progressive bradykinesia, dystonia, and other gait abnormalities [11,12,13]. There has been increasing concern regarding the role of environmental manganese exposure and children’s health [14]. Manganese has been identified as a risk factor for the development of aggressive behavior, attention deficit, cognitive decline resulting in lowered IQ, and learning deficits in infants and children [15,16,17,18].

Analysis of brain samples have shown that manganese accumulates within the human striatum, globus pallidus, and substantia nigra [3,19]. Manganese accumulation in these brain regions is associated with the presence of the divalent metal transporter 1 (DMT1) although additional transporters may play a role in brain uptake of manganese [20,21,22]. Brain imaging studies that rely on the paramagnetic properties of manganese that result in increased signal intensity seen with T1-weighted magnetic resonance imaging (MRI), allow for visual inspection of the brain for evidence of manganese accumulation at this site. Brain MRI studies of highly exposed people reveal signal intensity changes in the globus pallidus, striatum, and midbrain consistent with manganese accumulation at these sites [23,24]. Studies performed in nonhuman primates have shown that changes in the T1-weighted image correlate with manganese tissue concentration [25]. The primary neuropathologic target of manganese neurotoxicity is the globus pallidus (particularly the internal segment) with sparing of the substantia nigra pars compacta and an absence of Lewy bodies [26]. Studies of a manganese-exposed South African mine worker have revealed reduced astrocyte and neuron density in both the caudate and putamen [27]. Chronic manganese neurotoxicity in people is also associated with decreased γ-aminobutyric acid (GABA) neurons, reduced myelinated fibers, and moderate astrocytic proliferation in the medial segment of the globus pallidus [26].

Several studies have examined neurochemical changes following high-dose-manganese exposure. Because manganese neurotoxicity results in dysregulation of motor control, many studies have focused on the striatal and pallidal dopaminergic system. Manganese reacts with dopamine and other biogenic amines resulting in oxidative damage to the neurotransmitters [28]. One pathway involves manganese catalyzed oxidation of the alpha hydroxyl group of dopamine forming a semi-quinone radical (Figure 1). The semi-quinone radical then reacts with oxygen to generate superoxide anion radical [O_2_•−] and a quinone. Oxygen can reoxidize the quinone to quinol to generate hydrogen peroxide [28]. Manganese-catalyzed dopamine auto-oxidation may also involve semiquinone and aminochrome intermediates, l-cysteine or copper, and NADH facilitation [29,30]. Excess manganese may also alter glutamate homeostasis in the basal ganglia [31]. Changes in glutamate homeostasis have been associated with excitotoxicity in the CNS [31].

In vivo mammalian studies have shown that manganese exposure can result in altered levels of dopamine and its metabolites (e.g., 3,4-dihydroxyphenylacetic acid) and alter dopaminergic neurotransmission [32,33]. Alterations in manganese dopamine transmission can occur in the absence of detectable neuropathology. Although the initial focus on neurochemical effects is primarily centered on dopaminergic transmission, alterations in other neurotransmitter systems, including GABA and acetylcholine, also occur following manganese exposure [34,35]. Changes in striatal GABA, norepinephrine, and serotonin function are seen following manganese exposure in rodents [35].

Important species differences have been demonstrated with respect to manganese neurotoxicity [3]. Rodents generally fail to develop a behavioral syndrome or neuropathological lesion comparable to that seen in manganese-poisoned humans. Nevertheless, experimental studies have demonstrated that high-dose-intraperitoneal-manganese exposure impairs memory, learning ability and locomotor function in mice [36,37]. Manganese exposure also results in dopaminergic neuron loss in the striatum of MitoPark mice, a mitochondrially defective transgenic mouse model of Parkinson’s Disease [38]. Nonhuman primates best replicate the neurotoxic effects observed in humans. Manganese-exposed monkeys develop gait and other motor abnormalities that mimic those observed in affected humans [39,40,41]. Monkeys also develop reduced levels of striatal and pallidal dopamine and 3,4-dihydroxyphenylacetic acid, in conjunction with loss of dopaminergic neurons. These changes occur in the absence of loss of dopamine terminals in the caudate and putamen [39]. Monkeys also develop deficits in spatial and non-spatial working memory as well as effects on visuospatial-paired associate learning [42,43,44]. Histological assessment of the frontal cortex from manganese-exposed monkeys has also shown the presence of cells with apoptotic stigmata and astrocytosis in both the gray and white matter and α-synuclein aggregation in the frontal cortex gray and white matter [45].

The mechanism of action of manganese neurotoxicity remains the subject of ongoing research [46]. Molecular mechanisms of manganese neurotoxicity involve multiple neuronal cell types and can include mitochondrial impairment, oxidative stress, inflammation, and excitotoxicity [46,47]. Manganese can trigger glial activation and neuroinflammation in both microglia and astrocytes [48]. Welders exposed to manganese have altered methylation patterns in the DNA that codes for inducible nitric oxide synthase [49]. This review largely focuses on the role of oxidative stress in manganese neurotoxicity (Figure 2) with special attention to the role of work from the Aschner laboratory.

## 2. Reactive Oxygen Species (ROS) and Neurotoxicity

Reactive oxygen species (ROS) are molecular oxygen products that play an important role in normal biology and disease [50]. Both non-radical (e.g., hydrogen peroxide, molecular oxygen) and free radical (e.g., superoxide anion radical [O_2_•−], hydroxyl radical [•OH], peroxyl radical [ROO•]) forms of ROS exist. Normal endogenous production of cellular ROS primarily occurs through mitochondrial oxidative phosphorylation at the electron transport chain when molecular oxygen is reduced [51]. During transport, some electrons react with molecular oxygen forming O_2_^−^• [52]. Elevated mitochondrial ROS production can cause mitochondrial dysfunction and contribute to neurologic disease [53,54].

Despite its role in disease, ROS are also required for numerous normal cellular processes including cell growth, differentiation, and death by acting as signaling molecules. ROS activates the nuclear factor erythroid 2 (NF-E2)-related factor 2/Kelch-like ECH-associated protein 1 (NRF2/KEAP1) pathway, which serves as a master regulator of ROS levels [55,56]. Other pathways regulated by ROS include nuclear factor-κB (NF-κB), phosphoinositide 3-kinase (PI3K)/AKT, and mitogen-activated protein kinase (MAPK) [57]. As a result, ROS activate diverse molecular targets, initiating pathways involved in growth promotion and survival (including autophagy) or apoptosis [58].

Redox homeostasis within cells is the result of cellular processes that balance ROS production and antioxidant capacity. Oxidative stress occurs when this balance is perturbed and ROS production overwhelms the cellular antioxidant defense, damaging nucleic acids, proteins, and lipids. The brain has several features that predispose it to oxidative stress, including disproportionately high-oxygen consumption, lack of stored ATP, high-lipid content, among others [28]. There is growing literature that oxidative stress plays a role in manganese neurotoxicity as well as other neurodegenerative diseases, including Alzheimer’s disease, Parkinson’s disease (PD), Huntington’s disease (HD), and aging [59].

## 3. Oxidative Stress and Manganese: Physicochemical Properties

Manganese has eleven oxidation states ranging from Mn^−3^ to Mn^+7^, with Mn^+7^ having the strongest oxidation state of the group [60]. In the body, manganese exists in several valence states including divalent (Mn^2+^) and trivalent (Mn^3+^) forms. The reduced form of manganese (Mn^+2^) has a lower oxidative stress potential when compared with the trivalent (Mn^+3^) oxidized form of this metal [61,62,63]. The valence form of manganese can influence in vitro cytotoxicity with the trivalent form being more potent than the divalent form [63,64]. Intracellular formation of Mn^3+^, following oxidation of Mn^2+^, likely plays a minimal direct role in manganese neurotoxicity [65]. Manganese can secondarily alter the redox balance of iron, copper, and other transition metals [60,62,66]. Manganese promotes mitochondrial peroxide (H_2_O_2_) production even at physiologic concentrations [67]. Given the ability of manganese to participate in redox reactions, it is widely hypothesized that oxidative stress plays a role in manganese neurotoxicity [68].

In biological systems the redox active transition metals iron and copper, may participate in electron transfer reactions that produce hydroxyl free radical (·OH) via Fenton-type reactions resulting in oxygen radical damage [69,70]. Evolutionarily cells have developed protective antioxidant systems to scavenge these reactive oxygen products. The higher reduction potential of manganese when compared with iron limits its participation in Fenton-like redox chemistry [71].

## 4. Anti-Oxidants as Mediators of Manganese Neurotoxicity

Relative to the liver, the brain has lower levels of catalase activity, cytosolic GSH concentrations, and glutathione peroxidase 4 expression [28]. Reduced glutathione peroxidase 4 expression has been linked with ferroptosis, an intracellular iron-dependent form of cell death that has been proposed as a contributing mechanism in manganese neurotoxicity [72]. Another family of proteins with antioxidant properties are the peroxiredoxins. These peroxidases are involved in redox homeostasis, phospholipid turnover, glycolipid metabolism, and cellular signaling [73]. Peroxiredoxin 2 is present in the central nervous system and reduces ROS production by catalyzing hydrogen peroxide [74]. Metallothionein also plays a role as a free radical scavenger and is also involved in the metabolism of zinc and some other metals [75,76]

Manganese is incorporated into manganese superoxide dismutase (MnSOD) the principal antioxidant enzyme found in mammalian cells. This enzyme converts superoxide anion radicals to hydrogen peroxide and oxygen in mitochondria [77]. Thus, MnSOD plays a critical role in mitochondrial and cellular redox homeostasis and protects cells from oxidative stress. Genetic deletion of MnSOD is typically lethal in rodents, while a neuron specific deletion in spinal cord neurons results in extensive demyelination and axonal degeneration, elevated production of inflammatory cytokines, and microglia activation [78].

Additional discussion of the role of antioxidants in manganese neurotoxicity follows.

## 5. The Role of Mitochondrial Oxidative Stress in Manganese Neurotoxicity

The human brain’s reliance on ATP production leads to it consuming approximately 20% of the total basal oxygen budget [28]. Meeting neuronal ATP demands requires mitochondria, which in addition to ATP production are also involved in cell signaling, calcium homeostasis, and other cellular processes. Mitochondria are also the main intracellular storage site for manganese [79]. Manganese is mainly transported into the mitochondria via the mitochondrial Ca^2+^ uniporter system [80]. As mentioned earlier, mitochondrial manganese primarily exists as Mn^2+^ [79]. Although Mn^+2^ is less reactive when compared with Mn^+3^, increased mitochondrial manganese concentrations have numerous biological effects, including inhibition of oxidative phosphorylation [81,82], increased mitochondrial matrix calcium concentration [83], and inhibition of brain mitochondria respiratory complexes I–IV [84,85].

Altered mitochondrial activity in rat pheochromocytoma (PC12) cells treated with manganese is associated with altered mitochondrial activity [85]. Changes in mitochondrial function in the PC12 cells occurred along with reduced glutathione (GSH) concentrations and decreased catalase activity [85]. These effects can result in increased production of reactive oxygen (ROS) and reactive nitrogen species with subsequent oxidative stress. Pretreatment of cell cultures with antioxidants (e.g., ascorbic acid, GSH, N-acetyl cysteine) can mitigate some manganese-induced effects on mitochondrial function [86,87,88]. Manganese-induced impairment of mitochondrial membrane potential is partially rescued by pretreatment with inhibitors of p53 transcriptional activity and p53 mitochondrial translocation [89]. Rat striatal neurons develop dose-dependent decreases in mitochondrial membrane potential and complex II activity following in vitro exposure to manganese [90]. Striatal neurons exposed for two days to manganese at 5 μM developed DNA fragmentation and decreased expression of microtubule-associated protein MAP-2, suggesting that manganese may trigger apoptotic-like neuronal death secondary to mitochondrial dysfunction [90]. However, recent studies have shown that manganese-induced effects on mitochondrial function only occur at concentrations that initiate cell death, suggesting that mitochondrial dysfunction plays a limited role in cytotoxicity [91].

The weight of evidence suggests that mitochondrial dysfunction may be an initiating event for manganese neurotoxicity leading to overproduction of both ROS and reactive nitrogen species (RNS). Altered ROS and RNS production results in altered cell signaling, including activation of proinflammatory signaling and apoptotic cell death [92].

## 6. *Caenorhabditis elegans* as an Animal Model of Manganese Neurotoxicity

There is a growing interest in toxicology in the use of so-called New Alternative Models (NAMs) to reduce reliance on mammalian-based toxicity testing [93]. The Aschner laboratory has been at the forefront of using the nematode, *C. elegans*, to study the roles of oxidative stress, mitochondrial dysfunction, and dopaminergic neurodegeneration following manganese exposure [94,95]. The approximately 19,000 genes in the genome of this nematode have 60–80% homology with the mammalian genome [96] and *C. elegans* and mammals share many biological functions. The *C. elegans* nervous system is completely defined with 302 neurons and 56 glial cells or 381 neurons and 92 glial cells in either hermafrodite or males, respectively [97,98]. The *C. elegans* hermaphrodite possesses eight dopaminergic neurons, consisting of three pairs within the head and one pair in a posterior lateral position [99,100]. The dopaminergic system of *C. elegans* is more sensitive to the effects of manganese when compared with other neuron classes [101]. Exposure of early stage (L1) *C. elegans* larvae to manganese results in degeneration of these dopaminergic neurons in L1, L4 and young adults [102,103,104,105]. Exposure of *C. elegans* to manganese also results in behavioral changes. For example, manganese exposure alters olfactory learning and memory in L1 *C. elegans* [106].

Manganese toxicity in *C. elegans* can result in reduced GSH levels, generation of ROS, mitochondrial changes, and death [105,107,108]. Recent studies investigating this association have used nematodes with mutant forms of hpo-9 (e.g., tm3719), the worm homolog of BTBD9. These worms demonstrate hyperactive egg-laying behavior and have been proposed as a model organism for the study of restless leg syndrome in people [109]. When compared with wild type nematodes, tm3719 and hpo-9 knockout worms were more sensitive to manganese exposure with higher production of ROS and decreased numbers of intact mitochondria [110].

## 7. Astrocytes as Targets for Manganese Neurotoxicity

Interactions between neurons and astrocytes are critical for maintaining homeostasis of glutamate, glutamine, and GABA. Astrocytes contain glutamine synthetase, which catalyzes the conversion of L-glutamate, ATP, and ammonia into L-glutamine, ADP, and phosphate. Synthesized glutamine is subsequently released extracellularly and taken up by neurons and metabolized to glutamate by glutaminase. Synaptic glutamate released from neurons is removed by astrocytes through several cell membrane sodium-dependent transporters, including glutamate-aspartate transporter (GLAST) and glutamate transporter 1 (GLT1).

Manganese is preferentially localized in astrocytes in the brain at levels that are 50–200 times higher than those seen in neurons [111,112]. Manganese influences astrocyte morphology [113,114]. In vitro studies have shown that astrocyte uptake of Mn^2+^ depends on transferrin and DMT1 [115]. Expression and activity of glutamine synthetase in rat primary astrocytes is reduced following in vitro exposure to manganese [116]. In vitro studies using Chinese hamster ovary cells with increased expression of GLAST and GLT-1 have shown that manganese reduces glutamate transport into these cells [117]. Manganese deregulates expression of glutamine and glutamate transporters via protein kinase C pathway activation [118,119]. Downregulation of GLAST and GLT-1 transporter expression also occurs in rhesus monkeys following manganese inhalation [120]. Treatment of astrocytes with either estrogen or tamoxifen will reverse manganese-induced glutamate transporter impairment in astrocytes via increased transforming growth factor beta1 expression [121].

In vitro treatment of astrocytes with manganese results in shifts in the intracellular redox potential towards the oxidized state and results in induction of oxidative stress, mitochondrial dysfunction, and altered glutamine/glutamate cycling [122,123]. Exposure of astrocytes to manganese increases nitric oxide (NO) synthesis in these glial cells [124]. Astrocytes treated with manganese also demonstrate enhanced expression of inflammatory cytokines and chemokines that can be amplified by neighboring microglial cells [125,126]. Manganese activates astrocyte caspase-3 and phosphorylated extracellular signal-regulated kinase (p-ERK) via mitochondrial-dependent pathways [127]. Down regulation of the redox sensing protein 1 (DJ-1)/PARK7 increases the susceptibility of astrocytes to manganese-induced oxidative stress [128].

Impacts of manganese on other glial cells should not be ignored. For example, NF-κB signaling in microglia regulate the production and release of cytokines and chemokines that amplify the activation of astrocytes [125,126,129]. Thus, microglial play a role in mediating neuroinflammatory responses during manganese neurotoxicity. Microglial cells are also involved in ROS production and oxidative stress. In vitro exposure of rat microglia to manganese results in a time- and concentration-dependent release of hydrogen peroxide [130]. Manganese-induced release of hydrogen sulfide by microglia was reduced by mitogen-activated protein kinases inhibitors. Manganese treatment of microglia also activated ERK and p38-MAPK that preceded hydrogen peroxide production [130]. Studies with microglia-depleted dopaminergic neuron cultures show that depletion of microglia reduces manganese-induced neuron injury [131].

## 8. The Role of Oxidative Stress in Manganese Neurotoxicity: In Vivo Mammalian Studies

My laboratory enjoyed a multi-year collaboration with the Aschner laboratory that evaluated markers of oxidative stress in rodents following manganese inhalation [132]. Several exposure paradigms were used in these inhalation studies including 14-day, subchronic, and developmental studies [133,134,135,136]. During the subchronic study [135], young adult male and female CD rats and senescent male rats were exposed 6 h/day, 5 days/week for 90 days to air or manganese sulfate at 0.01, 0.1, or 0.5 mg Mn/m^3^ or manganese phosphate at 0.1 mg Mn/m^3^. Oxidative stress biomarkers that were evaluated by Aschner and colleagues included measurement of GSH and metallothionein concentrations, and glutamine synthetase protein levels, as well as metallothionein and glutamine synthetase mRNA levels in the cerebellum, olfactory bulb, striatum, hippocampus, and hypothalamus from control and manganese-exposed rats [137,138,139,140]. Depletion of oxidative scavengers such as glutathione and metallothionein occurred in some brain regions [138]. In juvenile and 16-month-old rats that inhaled either manganese sulfate or manganese phosphate, total GSH levels significantly decreased in the olfactory bulb of manganese-exposed young males and increased in the female olfactory bulb. Both aged and young female rats had significantly reduced GSH in the striatum following manganese inhalation. Senescent male rats exhibited decreased GSH levels in the cerebellum and hypothalamus following manganese inhalation [140]. Total GSH level was reduced in several brain regions of male rats exposed to manganese during early development. In female rats, GSH was unchanged, or upregulated in the olfactory bulb [68].

Other in vivo rodent studies also report manganese-induced oxidative stress in various regions of the rodent brain, particularly in the basal ganglia, such as globus pallidus, striatum, and substantia nigra [34,61,141,142,143,144]. Conflicting data from rodent studies have also been reported. For example, a study from our laboratory failed to demonstrate evidence of increased striatal ROS or whole-brain 8-hydroxy-2’ -deoxyguanosine (8-OHdG) levels despite increased brain and mitochondrial manganese concentrations, altered motor activity, and decreased body weight in rats following either adult or developmental exposure to oral manganese [145].

As noted earlier, unlike rodents, nonhuman primates largely replicate the neurotoxic effects observed in humans [7]. Chronic manganese exposure did not produce the loss of dopamine terminals in the caudate and putamen based on [^11^C]- methylphenidate PET imaging of exposed animals [146,147]. Histologic assessment of the frontal cortex from manganese-exposed monkeys has revealed the presence of cells with apoptotic stigmata and astrocytosis in both the gray and white matter and a-synuclein aggregation in the frontal cortex gray and white matter [45]

The role of oxidative stress in manganese neurotoxicity in nonhuman primates has also been examined and largely stems from studies performed in our laboratory to characterize manganese pharmacokinetics in young male rhesus monkeys following inhalation [148]. In this study, monkeys were exposed to either air or manganese sulfate at either 0.06, 0.3, or 1.5 mg Mn/m^3^ for 65 exposures. Additional monkeys were exposed to manganese sulfate at 1.5 mg Mn/m^3^ for 15 or 33 exposures and evaluated immediately thereafter or for 65 exposures followed by a 45- or 90-day delay before evaluation [148]. Brain imaging studies using T1-weighted magnetic resonance imaging revealed dose-dependent increases in MRI signal hyperintensities within the olfactory bulb and the globus pallidus [25]. As the exposure increased, manganese-induced hyperintensities involved multiple brain regions that were confirmed using chemical analysis of affected bran regions.

Biochemical endpoints indicative of oxidative stress and excitotoxicity were assessed in the cerebellum, frontal cortex, caudate, globus pallidus, olfactory cortex, and putamen of monkeys exposed to manganese [120,149]. Glutamine synthetase, GLT-1, GLAST and tyrosine protein levels, metallothionein, GLT-1, GLAST, tyrosine hydroxylase and GS mRNA levels, and total GSH levels were determined for all brain regions [120,149]. Manganese exposure differentially affected these biomarkers in each brain region. For example, GSH was increased in the frontal cortex and decreased in the caudate despite two- to threefold increases in manganese concentrations in these regions. Exposure to manganese sulfate persistently decreased metallothionein mRNA in the caudate when compared to air-exposed controls. In contrast, putamen metallothionein mRNA levels were unaffected by manganese exposure. The glutamate transporters GLT-1 and GLAST were relatively unaffected by short term manganese exposure, except in the globus pallidus where exposure for 33 days led to decreased protein levels, which persisted after 45 days of recovery for both proteins and 90 days of recovery in the case of GLAST. Exposure to 1.5 mg Mn/m^3^ caused a significant decrease in GSH levels in the caudate and increased GSH levels in the putamen of monkey exposed for 15 and 33 days, with both effects persisting at least 90 days post-exposure. Tyrosine hydroxylase protein levels were significantly lowered in the globus pallidus of the monkeys exposed for 33 days, but mRNA levels were significantly increased in this same region. All manganese-exposed monkeys had decreased pallidal glutamine synthetase protein, decreased caudate GLT-1 mRNA, decreased pallidal GLAST protein, and increased olfactory tyrosine hydroxylase mRNA levels.

Studies exploring the role of oxidative stress in human manganese neurotoxicity are extremely limited and primarily rely on in vitro exposure of human cells, as reviewed earlier. Additional studies using noninvasive methods to assess redox status in manganese-exposed humans are needed.

## 9. Conclusions

As this review demonstrates, molecular mechanisms are involved with manganese neurotoxicity. Although oxidative stress may play a critical role, it may not serve as the initiating event. Rather it is likely that the initiating event is an increase in intracellular levels of manganese because of manganese overexposure. Pharmacokinetic models predict that increases in globus pallidal manganese concentrations in humans above approximately 0.55 μg Mn/g trigger adverse neurologic effects [150]. Elevated brain manganese levels are partially sequestered in mitochondria, resulting in mitochondrial dysfunction and secondary oxidative stress. Elevated production of ROS can lead to further mitochondrial injury resulting in neurotoxicity. Thus, the respective roles of manganese overexposure, mitochondrial dysfunction, and oxidative stress are tightly interwoven.

This review has largely focused on only one aspect of research stemming from Professor Aschner’s laboratory. Work from Professor Aschner’s laboratory has provided a wealth of data demonstrating that oxidative stress and mitochondrial dysfunction are critical molecular mechanism involved in manganese neurotoxicity. His work has provided insights into how elevated intracellular manganese concentrations results in a cascade of events linking oxidative stress and mitochondrial dysfunction with abnormal cellular function. This research is not only relevant for manganese neurotoxicity but also promises to hold broader future impact in improving our understanding of mechanisms involved in other neurological disorders. Aschner’s work with *C. elegans* also helps pave the way for the use of this animal model in neurotoxicity research. This effort is well aligned with ongoing international efforts to decrease reliance on vertebrate animal models in neurotoxicology. Dr. Aschner’s work also helps to illustrate the power of collaborative science and serves as a model for other scientists.

## Figures and Tables

**Figure 1 biomolecules-13-01176-f001:**
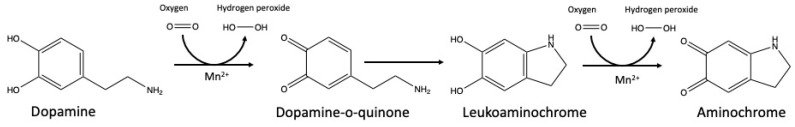
Manganese-catalyzed autoxidation of dopamine involves redox cycling of Mn^2+^ and Mn^3+^ in a series of reactions that generate hydrogen peroxide (H_2_O_2_), dopamine-*o*-quinone and aminochrome.

**Figure 2 biomolecules-13-01176-f002:**
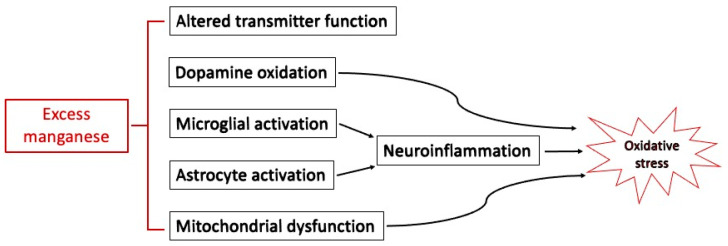
Mechanisms involved in manganese neurotoxicity.

## Data Availability

All data is contained herein.

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
