# Peer review of "The Role of Oxidative Stress in Manganese Neurotoxicity: A Literature Review Focused on Contributions Made by Professor Michael Aschner"

_biomolecules, 2023, doi:10.3390/biom13081176_

Round 1
Reviewer 1 Report
This is a nice, timely mini review of Dr Aschner's contributions to the field of toxicology. My only minor comment is that early in the review it is mentioned that the following will be focused on the Aschner studies yet it wasn't until section 5 that they were a focus. Until the C. Elegans work, it was a slightly confusing what was the work of Dr Aschner vs other laboratories.
Author Response
Thank you for the comments. I cite over 50 articles that stems from the Aschner laboratory. I highlighted the work on C. elegans since it represented some of the more recent efforts and in many ways some of the most unique contributions. I have not made changes to the manuscript in response to your comments.
Reviewer 2 Report
Manuscript ID: biomolecules-2510064
Type of manuscript: Review
Title: " The Role of Oxidative Stress in Manganese Neurotoxicity: A Literature Review Focused on Contributions Made by Professor Michael Aschner "
In this review the author aims to demonstrate the importance that Professor Michael Aschner's research has in the progression of studies involved with the neurotoxic effect of manganese, and simultaneously in increasing the understanding of the mechanisms involved.
As the author makes clear, the greatest emphasis is given to oxidative stress.
The author tries to address the various aspects related to this topic, having divided his review by a logic that seems somehow difficult to understand. However, he manages to address the most relevant aspects.
As the aim of this review is to show the contribution of Professor Aschner, although many studies are referred to, the main focus is on the work developed in his laboratory.
The important research that has been carried out with C. elegans, and its alignment with the decreasing use of vertebrate animals for neurotoxicity studies, is also mentioned.
At the end, the importance of collaborative work between different research groups is stressed, highlighting the importance that this aspect has played in the work developed by Professor Aschner's group.
In my opinion, this review is well written and well documented with bibliographic references and can be a good starting point for beginners in this area of research.
Minor corrections:
Line 30: 0.003 - 0.06 instead of "0.003 -0.6"
Line 289: delete "to exposed"
In my opinion, the review can be published in its current format in the Journal Biomolecules.
Author Response
Thanks for the comments. I have made the minor changes you have suggested:
Line 30: 0.003 - 0.06 instead of "0.003 -0.6" Note I deleted the infant data since it was not immediately cited in the reference.
Line 289: delete "to exposed": Corrected
In my opinion, the review can be published in its current format in the Journal Biomolecules. Thank you. I considered your comment regarding your concern regarding the logic I used to create the manuscript - I did not however make changes to the flow of the article since this did not appear to be a major concern.